# Molecular safeguarding of CRISPR gene drive experiments

Jackson Champer[1,2]*, Joan Chung[1,2], Yoo Lim Lee[1,2], Chen Liu[1,2], Emily Yang[1,2], Zhaoxin Wen[1], Andrew G Clark[1], Philipp W Messer[1]*

[1]Department of Biological Statistics and Computational Biology, Cornell University, Ithaca, United States; [2]Department of Molecular Biology and Genetics, Cornell University, Ithaca, United States

**Abstract** CRISPR-based homing gene drives have sparked both enthusiasm and deep concerns due to their potential for genetically altering entire species. This raises the question about our ability to prevent the unintended spread of such drives from the laboratory into a natural population. Here, we experimentally demonstrate the suitability of synthetic target site drives as well as split drives as flexible safeguarding strategies for gene drive experiments by showing that their performance closely resembles that of standard homing drives in *Drosophila melanogaster*. Using our split drive system, we further find that maternal deposition of both Cas9 and gRNA is required to form resistance alleles in the early embryo and that maternally-deposited Cas9 alone can power germline drive conversion in individuals that lack a genomic source of Cas9.
DOI: https://doi.org/10.7554/eLife.41439.001

*For correspondence:
jc3248@cornell.edu (JC);
messer@cornell.edu (PWM)

Competing interests: The authors declare that no competing interests exist.

## Introduction

Homing gene drives have the potential to rapidly spread through a population by converting wild-type alleles to drive alleles in the germline of heterozygous individuals, thereby enabling super-Mendelian inheritance of the drive allele (*Esvelt et al., 2014*; *Champer et al., 2016*; *Burt, 2014*; *Unckless et al., 2015*; *Alphey, 2014*; *Noble et al., 2017*; *Deredec et al., 2011*). Such systems could be a tool for the eradication of vector-borne diseases such as malaria or dengue by propagating transgenes through mosquito populations that prevent disease transmission (*Esvelt et al., 2014*; *Champer et al., 2016*; *Burt, 2014*; *Alphey, 2014*). Other proposed applications include the direct suppression of vector populations, invasive species, or agricultural pests (*Esvelt et al., 2014*; *Champer et al., 2016*; *Burt, 2014*; *Alphey, 2014*).

Proof-of-principle studies using CRISPR-based homing drive constructs have now been demonstrated in a variety of potential target systems. The first experiments to achieve successful drive conversion were conducted in yeast (*DiCarlo et al., 2015*; *Roggenkamp et al., 2018*; *Basgall et al., 2018*; *Shapiro et al., 2018*), flies (*Champer et al., 2018a*; *Oberhofer et al., 2018*; *KaramiNejadRanjbar et al., 2018*; *Gantz and Bier, 2015*; *Champer et al., 2017*), and mosquitoes (*Hammond et al., 2017*; *Hammond et al., 2016*; *Gantz et al., 2015*). These experiments revealed highly variable conversion efficiencies, ranging from close to 100% in *Saccharomyces cerevisiae* to between 19–62% in *Drosophila melanogaster* and 87–99% in *Anopheles*. Such variability could be due to several factors, including differences in the level and timing of Cas9 expression, the genomic targets, and organism-specific factors such as recombination rate. In *Anopheles*, for instance, conversion rates generally tend to be higher than in *D. melanogaster*, especially in males, consistent with the fact that there is no recombination in male flies. A recent study has further demonstrated successful drive conversion in mice (*Grunwald et al., 2019*), although with a lower efficiency than most of the other systems.

**eLife digest** Gene drives are a new genome editing technology where artificial gene packages are designed to create a mutation that will quickly spread within a population. These packages target a specific sequence in a genome, where they could potentially add, remove or deactivate a gene. They also trigger a process known as drive conversion, which ensures the mutation will be inherited at a higher rate than normal. Within several generations, nearly every organism in the population will carry this genetic change.

This technology could, for example, help us eradicate disease-carrying mosquitoes, crop pests or invasive species. However, it could also have unforeseen and dangerous consequences. It is therefore crucial to keep gene drives within laboratory walls before they are ready to be released. Even if a small numbers of genetically modified animals were to escape, they could rapidly spread the packages within a wild population.

To prevent this, scientists have devised two safeguarding strategies. One, called synthetic target site gene drive, uses target sequences that have been introduced on purpose in research organisms, but which are absent in wild populations. If the gene drive were to escape, it could not spread in the genomes of wild creatures because they lack the synthetic target site. Alternatively, split drive systems can also limit risk. There, the different components required for a gene drive are not packaged together, but in separate locations in the genome. Some of these elements are inherited at a normal rate, so the gene drive fizzles out after a few generations. However, it was still unclear whether synthetic gene drives and split drive systems could be used instead of the classic approach and yield the same results in research.

Champer et al. compared traditional gene drives, synthetic target site gene drives, and split drive systems in fruit flies raised in the laboratory. The experiments show that the three approaches lead to similar results, with the genetic package spreading and creating resistance in a similar way. They also confirm that, in split drive systems, both components of the drive must be genetically inherited to create the intended mutation.

Synthetic gene drives and split drive systems could therefore be used in experiments on gene drives, especially in studies with large numbers of organisms. Ultimately, adopting these measures could help to keep gene drive research safe, which may encourage more scientific teams to work on this technology and exploit its potential.

DOI: https://doi.org/10.7554/eLife.41439.002

It has also become clear that homing gene drives face a significant obstacle due to the frequent formation of resistance alleles when cleavage is repaired by end-joining, which typically generates mutations at the target site (*Champer et al., 2017*). This process has been observed to take place in the germline during failed drive conversion, but also in the embryo due to the persistence of maternally-deposited Cas9 (*Champer et al., 2017*). Similar to conversion rates, resistance rates too are highly variable between drive systems and organisms (*DiCarlo et al., 2015*; *Champer et al., 2017*; *Gantz et al., 2015*; *Grunwald et al., 2019*; *Champer et al., 2018b*; *Hammond et al., 2018*). However, strategies for improving conversion efficiency and lowering resistance rates have already been successfully tested, including gRNA multiplexing (*Champer et al., 2018b*), improved promoters (*Champer et al., 2018b*; *Hammond et al., 2018*), and careful selection of target sites to render resistance alleles non-viable (*Kyrou et al., 2018*). In fact, total population elimination with a CRISPR gene drive was recently achieved in laboratory cages of *Anopheles gambiae* for the first time (*Kyrou et al., 2018*).

While some have touted CRISPR homing drives as a potential game-changer in the fight against vector-borne diseases, key questions loom large about our ability to predict the outcome of releasing such a drive into a natural population. Unintended effects or even an accidental release could result in severe societal backlash. These concerns may seem hypothetical at present, given that most drives are still prone to rapid evolution of resistance (*KaramiNejadRanjbar et al., 2018*; *Champer et al., 2017*; *Unckless et al., 2017*). Yet even an inefficient drive that reaches only a modest fraction of the population may spread resistance alleles to the entire population (*Noble et al.,*

*2018*). Furthermore, the first examples of effective drives are already on the horizon (*Kyrou et al., 2018*), and even more powerful drives will likely be developed in the near future.

Regardless of the likelihood of their escape from a lab or a field trial, it is imperative that we safeguard laboratory gene drives so that they cannot accidentally spread into a natural population. Current strategies typically rely on physical confinement of drive-containing organisms. However, it is doubtful whether this sufficiently reduces the likelihood of any accidental escape into the wild given the possibility of human error. Since very few escapees can establish an effective drive in a population (*Unckless et al., 2015*; *Noble et al., 2018*; *Marshall and Hay, 2012*; *Marshall, 2009*), additional safety measures should be employed in any experiments with drives potentially capable of spreading indefinitely.

Two molecular safeguarding strategies have recently been proposed that go beyond physical or ecological confinement (*Akbari et al., 2015*). The first is synthetic target site drive, which homes into engineered genomic sites that are absent in the wild. The second is split drive, where the drive construct lacks its own endonuclease, relying on one engineered into an unlinked site instead. Both strategies should thereby reliably prevent efficient drive outside of their respective laboratory lines.

One potential drawback of these strategies is that each requires an additional transgenesis step compared to a standard drive. For a split drive, the line containing the Cas9 gene needs to be engineered, although one such line could be used for multiple split drive systems, and the transformation of the two individual elements may be easier since each is smaller than a standard drive. For the synthetic target site drive, the line containing the synthetic target needs to be separately engineered. However, such a system can also provide distinct advantages over standard drives in addition to confinement. For example, moving a target gene from a pest species into a model organism would permit researchers to test some aspects of the drive system in the model organism prior to release in the pest population. Additionally, the flexibility of synthetic target site drives allows targeting a dominant marker such as a fluorescent gene, facilitating the measurement of drive performance parameters while preventing the need to target a natural marker gene that may have significant fitness effects.

Here, we provide the first experimental demonstration of synthetic target site drives and split drives in an insect system and show that their behavior closely resembles that of standard drives, with similar rates of drive conversion efficiency and resistance allele formation. This suggests that these strategies can serve as appropriate molecular safeguards in the development and testing of CRISPR homing gene drives.

## Results and discussion

### Synthetic target sites

We designed and tested three synthetic target site drives in *D. melanogaster*, each targeting an enhanced green fluorescent protein (EGFP) gene introduced at two autosomal sites and an X-linked site adjacent to the *yellow* gene (*Figure 1a*). To determine conversion efficiencies of these drives (the percentage of EGFP alleles converted to drive alleles in the germline), we scored dsRed phenotype in the progeny of crosses between EGFP homozygotes and drive/EGFP heterozygotes. We found drive conversion efficiencies of approximately 52–54% in females and 32–46% in males in these drive/EGFP heterozygotes (*Table 1*, *Supplementary file 2-Datasets S1-S3*), which were similar to our previous homing drives targeting natural sites (*Champer et al., 2017*; *Champer et al., 2018b*). We next measured the rate at which 'r2' resistance alleles (those that disrupt the target gene) were formed in the embryo by scoring the progeny of female heterozygotes for EGFP phenotype. This rate was high in all three drives, ranging from 80 to 91% (*Table 1*, *Supplementary file 2-Datasets S1-S3*). It is thus likely that nearly all EGFP target alleles were converted to resistance alleles. These rates are again similar to what we found for previous drive constructs targeting the autosomal *cinnabar* and X-linked *white* loci (*Champer et al., 2018b*), but significantly less than a construct targeting the X-linked *yellow* (*Champer et al., 2017*) gene (p < 0.001, Fisher's exact test). This difference is likely due to location-specific variation in expression levels of Cas9 (and possibly also the gRNA) between constructs inserted into the *yellow* gene and other sites.

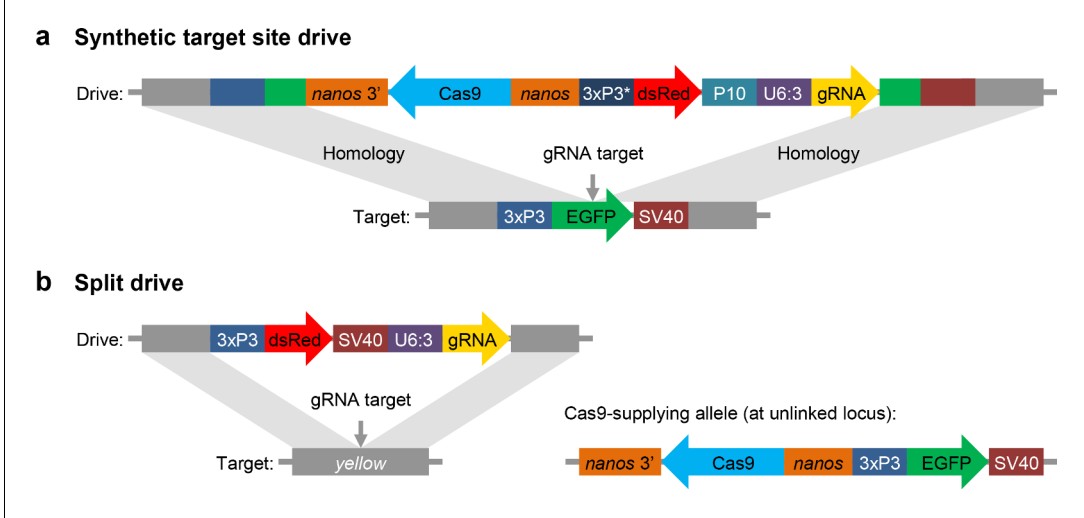

**Figure 1.** Schematic diagram of our synthetic target site drive and split drive constructs. (a) The synthetic target site drive constructs contain Cas9 with the germline *nanos* promoter and 3'UTR, a dsRed marker with a slightly recoded (*) 3xP3 promoter and P10 3'UTR, and a gRNA driven by the U6:3 promoter that targets the synthetic EGFP gene. The two homology arms include the EGFP sequence with its 3xP3 promoter and SV40 3'UTR regions. (b) The split drive contains a dsRed marker gene driven by a 3xP3 promoter together with a SV40 3'UTR, a gRNA expressed by the U6:3 promoter that targets *yellow*, and two homology arms for *yellow*. The unlinked supporting element contains Cas9 driven by the *nanos* promoter with a *nanos* 3'UTR, and an EGFP marker gene driven by a 3xP3 promoter together with a SV40 3'UTR.
DOI: https://doi.org/10.7554/eLife.41439.003

## Split drives

For our split drive system, we designed a drive construct targeting the X-linked *yellow* gene, similar to the one used in a previous study (*Champer et al., 2017*), but lacking Cas9 (*Figure 1b*). We then designed a second construct containing Cas9 driven by a *nanos* promoter for germline-restricted expression, which was inserted into chromosome 2R. We assessed drive performance of this system by first crossing males that had the drive element but no Cas9 to females that were homozygous for Cas9 but lacked the drive element. Similarly, we also crossed females homozygous for the drive but lacking Cas9 to males homozygous for Cas9 but lacking the drive. The progeny of these crosses followed Mendelian inheritance rules, indicating that both Cas9 and gRNA must be maternally deposited for resistance alleles to form in the early embryo.

The progeny of $w^{1118}$ males and drive/wild-type heterozygous females containing one copy of Cas9 were then scored for dsRed and yellow phenotype to assess drive conversion efficiency and resistance rates (*Table 1*, *Supplementary file 2-Datasets S4*). Compared to our previous results with a standard gene drive targeting *yellow* (*Champer et al., 2017*), where drive conversion efficiency was 62% (and which shared the same genomic location, gRNA, dsRed marker, and *nanos*-

**Table 1.** Drive performances of synthetic target site and split drives compared with the standard drives from our previous studies (*Champer et al., 2017*; *Champer et al., 2018b*).

| Drive | Male drive conversion efficiency | Female drive conversion efficiency | Embryo r2 resistance rate |
|---|---|---|---|
| EGFP site B | 32 ± 3% | 52 ± 3% | 88 ± 1% |
| EGFP site E | 46 ± 4% | 54 ± 5% | 91 ± 2% |
| EGFP site Y | N/A | 53 ± 3% | 80 ± 2% |
| *cinnabar* | 39 ± 3% | 54 ± 4% | 100 ± 0% |
| *white* | N/A | 59 ± 2% | 77 ± 2% |
| Split-*yellow* | N/A | 74 ± 2% | 74 ± 2% |
| *yellow* | N/A | 63 ± 3% | 20 ± 2% |

DOI: https://doi.org/10.7554/eLife.41439.004

Cas9 element, albeit at a different location), we measured a significantly higher drive conversion efficiency of 74% for the split drive using the same experimental parameters (p<0.0001, Fisher's exact test). This improvement may be due to increased efficiency of homology-directed repair for the split drive element compared to the larger standard drive. However, we also observed that early embryo r2 resistance allele formation was much higher in the split drive at 74% compared to the 20% for the standard drive (p<0.001, Fisher's exact test). This is likely because Cas9, rather than the gRNA, is the main limiting factor in determining the cleavage rate and that Cas9 at its new site had higher expression than the Cas9 in the standard drive at *yellow* (it was located only 277 nucleotides away from synthetic target site B, and the drive at this site had 88% embryo resistance).

One concern regarding the use of split drives as a surrogate for standard drives is that every genome in the experimental split drive population would contain Cas9, so maternally-deposited Cas9 would likely be present in each embryo, even if the mother did not have a drive element. In combination with the zygotically expressed gRNA from a paternal allele, this might then result in a higher rate of embryo resistance allele formation. However, our finding that both Cas9 and gRNA must be maternally deposited to form such embryo resistance alleles suggests that a split drive in a laboratory population should behave similarly to a standard drive.

A hypothetical split drive where Cas9 is encoded in the driving element and the gRNA forms the supporting element (the reverse of our split drive) would presumably have nearly identical behavior to a standard drive. This is because in such a drive the Cas9 gene would always be present in the same copy number per individual as in a standard drive, and it would be located at the same genomic position, eliminating the possibility of position-based differences in Cas9 expression levels between the two drives. It would also be much closer to the standard drive in total size, minimizing potential differences in the efficiency of homology-directed repair. However, such a strategy would be experimentally less flexible because both elements would have to be redesigned for every new target site, rather than just the drive element when Cas9 is in the supporting element. Nevertheless, such a strategy may be advantageous for testing standard homing drives, integral homing drives (*Nash et al., 2019*) with gRNAs driving in a separate synthetic target site, and other future types of CRISPR-based gene drives.

## Split drive mechanisms

The flexibility of the split drive system, facilitated by its genomic separation of Cas9 and gRNA, allowed us to further refine our understanding of the general mechanisms by which homing drives operate. Previous studies have indicated that germline resistance alleles can form in pre-gonial germline cells (*KaramiNejadRanjbar et al., 2018*; *Champer et al., 2018b*), but it remained unclear whether this could also occur at other stages. The fact that we observed a higher drive conversion efficiency of the split drive compared with a standard drive strongly implies that not all resistance alleles form prior to drive conversion, since resistance allele formation alone should not be affected by the reduced size of the split drive. This raises the possibility that drive conversion could potentially take place as an alternative to resistance allele formation in pre-gonial germline cells, where resistance alleles are known to form. However, a perhaps more likely explanation would be that only a portion of resistance alleles form in pre-gonial germline cells and the remainder form either in gametocytes as an alternative to drive conversion or afterward in late meiosis, when a template for homology-directed repair is no longer available.

We further found evidence that even in individuals lacking a genomic source of Cas9, maternally-deposited Cas9 can persist through to gametocytes in the germline, where it can then facilitate successful drive conversion. This was demonstrated by crosses of $w^{1118}$ males with females that were mosaic yellow and had inherited a drive allele with dsRed from their mother but not the Cas9 allele itself, as evidenced by the absence of a EGFP phenotype. These females still received maternally-deposited Cas9 from their heterozygous mothers. Despite lacking a Cas9 gene, fifteen out of sixteen of these flies showed an average drive conversion efficiency of 54%, with a single fly showing no drive conversion (*Supplementary file 2-Datasets S4*). By contrast, nine out of eleven females from the same cross that were fully yellow, rather than mosaic, and two wild-type females showed no detectable drive conversion, while two fully yellow females showed successful drive conversion. In the wild-type females, 'r1' resistance alleles that preserve the function of the target gene had presumably formed at the embryo stage. It is unlikely that any drive conversion had occurred in these females with full yellow phenotype at the early embryo stage, because in that case their progeny

should have consistently displayed biased inheritance of the drive allele. Thus, it appears that when Cas9 and gRNA are maternally-deposited, they can fail to cleave the target site in the early embryo and induce end-joining repair, while nonetheless showing significant cleavage activity in later stages when homology-directed repair is possible. This creates yellow phenotype over most of their body while also usually enabling drive conversion in the germline. Such Cas9 did not persist to embryos of the subsequent generation, as indicated by the lack of yellow phenotype in female progeny.

We also found that maternally-deposited gRNA was not necessary to achieve drive conversion in conjunction with maternally-deposited Cas9 when a genomic source of gRNA is provided. For example, we found that the progeny of drive-heterozygous females receiving a paternal drive allele without genomic Cas9 (but receiving maternally-deposited Cas9 from a mother with a single copy of Cas9) showed 38% germline drive conversion efficiency (*Supplementary file 2-Datasets S4*). An additional 12% of wild-type alleles were converted to r2 resistance alleles, and the remainder most likely remained wild-type. These rates were lower than those of the full split drive, likely because of reduced Cas9 and gRNA activity due to the fact that only maternally-deposited Cas9 could be utilized.

Taken together, our results suggest that in the absence of early embryo resistance alleles, germline drive rates in an individual may be affected by the level of maternally-deposited Cas9 and gRNAs. Individuals inheriting a drive allele from their mother, as opposed to their father, will also receive maternally-deposited Cas9, which could increase the level of cleavage during the window for homology-directed repair, thereby increasing the rate of drive conversion. On the other hand, cleavage by the maternally-deposited Cas9 prior to this stage in pre-gonial germline cells, in addition to the early embryo, should usually form additional resistance alleles compared to individuals that have no such persistent Cas9.

## Conclusions

Our results demonstrate that CRISPR homing gene drives with synthetic target sites such as EGFP will show highly similar behavior to standard drives and can thus be used for most testing in lieu of these drives. Split drives also show similar performance, while allowing for the targeting of natural sequences in situations where the use of synthetic targets is difficult, such as for certain resistance reduction strategies and population suppression drives that require the targeting of wild-type genes.

We therefore suggest that gene drive research should consistently adopt these molecular safeguarding strategies in the development and testing of new drives. This will be particularly important for large-scale cage experiments aimed at gaining a better understanding of the expected population dynamics of candidate drives, which will be integral for any informed discussion about their feasibility and risks.

## Materials and methods

### Design of the synthetic target site drives

We constructed three synthetic target site drives at different genomic sites (B, E, and Y) into which the EGFP target was inserted. Sites B and E were on chromosomes 2R and 3R, respectively, located 3' of two protein- coding genes. Site Y was on the X chromosome immediately downstream of *yellow*. All of our synthetic target site drives used a slightly recoded 3xP3 promoter (3xP3v2) to drive the dsRed marker and also used a P10 3'UTR. This was to reduce potential misalignment with the 3xP3 promoter and SV40 3'UTR in the homology arms (see *Figure 1a*), which we found to result in poor drive efficiency in initial tests of drive constructs that used the same 3xP3 promoter and SV40 3'UTR for the EGFP and dsRed markers.

### Genotypes and phenotypes

Since all our synthetic target site drives home into EGFP, successful insertion of the drives will disrupt this marker (*Figure 1a*). For the split drive, the driving element disrupts *yellow*, causing a recessive yellow body phenotype (*Figure 1b*). If cleavage is repaired by end-joining, rather than homology-directed repair, this will typically result in a mutated target site, creating a resistance allele. Most such resistance alleles will render the target gene nonfunctional due to a frameshift or

otherwise sufficient change in the amino acid sequence. We term these alleles 'r2'. Resistance alleles that preserve the function of the target gene are termed 'r1'. In some cases, we observed mosaicism for EGFP in the eyes of heterozygotes for the drive and the synthetic target site. This indicates that the germline *nanos* promoter may still drive a low level of expression in somatic cells. However, since no mosaicism was observed in the body of the split drive flies, this mosaicism may be due to proximity of the *nanos* promoter to the nearby 3xP3 promoter that drives expression in the eyes (the promoters were only eight nucleotides apart in the synthetic target site flies but 68 nucleotides apart in the split Cas9 construct). The different phenotypes and genotypes of our drive systems are summarized in *Supplementary file 2-Datasets S1-S3*, as are calculations for determining drive performance parameters based on phenotype counts.

## Generation of transgenic lines

One line in the study was transformed at GenetiVision by injecting the donor plasmid (ATSabG) into a $w^{1118}$ *D. melanogaster* line, and seven lines were transformed at Rainbow Transgenic Flies by injecting the donor plasmid (ATSaeG, ATSxyG, BHDgN1bv2, BHDgN1e, BHDgN1y, BHDaaN, IHDyi2) into the same $w^{1118}$ line. Cas9 from plasmid pHsp70-Cas9 (*Gratz et al., 2013*) (provided by Melissa Harrison and Kate O'Connor-Giles and Jill Wildonger, Addgene plasmid #45945) and gRNA from plasmids BHDaag1, BHDabg1, BHDaeg1, or BHDxyg1 were included in the injection, depending on the target site. For Genetivision injections, concentrations of donor, Cas9, and gRNA plasmids were 102, 88, and 60 ng/µL, respectively in 10 mM Tris-HCl, 23 µM EDTA, pH 8.1 solution. For Rainbow Transgenic Flies injections, concentrations of donor, Cas9, and gRNA plasmids were approximately 350–500, 250–500, and 50–100 ng/µL, respectively in 10 mM Tris-HCl, 100 µM EDTA, pH 8.5 solution. Note that the synthetic target site drives were transformed into the $w^{1118}$ line in parallel with the synthetic targets themselves, including elements of the target on either side of the drive. This avoids the need to transform the drive into lines already possessing the synthetic target site. To obtain homozygous lines, the injected embryos were reared and crossed with $w^{1118}$ flies. The progeny with dsRed or EGFP fluorescent protein in the eyes, which usually indicated successful insertion of the donor plasmid, were selected and crossed with each other for several generations. The stock was considered homozygous at the drive locus after sequencing confirmed lack of wild-type or resistance alleles.

## Fly rearing and phenotyping

All flies were reared at 25 °C with a 14/10 hr day/night cycle. Bloomington Standard medium was provided as food every 2–3 weeks. During phenotyping, flies were anesthetized with $CO_2$ and examined with a stereo dissecting microscope. Flies were considered 'mosaic' if any discernible mixture of green fluorescence was observed in either eye. However, for the synthetic target site drives, flies that carried a drive allele were only considered mosaic if either eye had less than 50% EGFP phenotype coverage, to avoid identifying flies with possible somatic expression of Cas9 as mosaic for EGFP. This definition was stringent enough that no mosaic insects without the drive were found that would have avoided mosaic classification based on this definition. Fluorescent phenotypes were scored using the NIGHTSEA system only in the eyes (SFA-GR for dsRed and SFA-RB-GO for EGFP). Even though dsRed did bleed through into the EGFP channel, both types of fluorescence could still be easily distinguished.

All experiments involving live gene drive flies were carried out using Arthropod Containment Level two protocols at the Sarkaria Arthropod Research Laboratory at Cornell University, a quarantine facility constructed to comply with containment standards developed by USDA APHIS. Additional safety protocols regarding insect handling approved by the Institutional Biosafety Committee at Cornell University were strictly obeyed throughout the study, further minimizing the risk of accidental release of transgenic flies.

## Genotyping

To obtain the DNA sequences of gRNA target sites, individual flies were first frozen and then ground in 30 µL of 10 mM Tris-HCl pH 8, 1 mM EDTA, 25 mM NaCl, and 200 µg/mL recombinant proteinase K (Thermo Scientific). The homogenized mixture was incubated at 37 °C for 30 min and then 95 °C for 5 min. 1 µL of the supernatant was used as the template for PCR to amplify the gRNA target site.

DNA was further purified by gel extraction and Sanger sequenced. Sequences were analyzed using the ApE software, available at: http://biologylabs.utah.edu/jorgensen/wayned/ape.

## Plasmid construction

The starting plasmid pCFD3-dU6:3gRNA (*Port et al., 2014*) (Addgene plasmid #49410) was kindly supplied by Simon Bullock, starting plasmid pJFRC81-10XUAS-IVS-Syn21-GFP-p10 (*Pfeiffer et al., 2012*) was a gift from Gerald Rubin (Addgene plasmid # 36432), and starting plasmid IHDyi2 was constructed in our previous study (*Champer et al., 2017*). All plasmids were digested with restriction enzymes from New England Biolabs (HF versions, when available). PCR was conducted with Q5 Hot Start DNA Polymerase (New England Biolabs) using DNA oligos and gBlocks from Integrated DNA Technologies. Gibson assembly of plasmids was conducted with Assembly Master Mix (New England Biolabs) and plasmids were transformed into JM109 competent cells (Zymo Research). Plasmids used for injection into eggs were purified with ZymoPure Midiprep kit (Zymo Research). Cas9 gRNA target sequences were identified using CRISPR Optimal Target Finder (*Gratz et al., 2014*). Tables of the DNA fragments used for Gibson Assembly of each plasmid, the PCR products with the oligonucleotide primer pair used, and plasmid digests with the restriction enzymes are shown in the Supporting Information.

## Acknowledgements

We thank Kevin Esvelt, Owain Edwards, and one anonymous reviewer for their constructive comments that helped improve the manuscript. This study was supported by startup funds from the College of Agriculture and Life Sciences at Cornell University to PWM, the National Institutes of Health award R21AI130635 to JC, AGC, and PWM, and the National Institutes of Health award F32AI138476 to JC.

## Additional information

### Funding

| Funder | Grant reference number | Author |
| --- | --- | --- |
| National Institutes of Health | F32AI138476 | Jackson Champer |
| National Institutes of Health | R21AI130635 | Jackson Champer Andrew G Clark Philipp W Messer |

The funders had no role in study design, data collection and interpretation, or the decision to submit the work for publication.

### Author contributions

Jackson Champer, Conceptualization, Data curation, Formal analysis, Supervision, Funding acquisition, Validation, Investigation, Visualization, Methodology, Writing—original draft, Writing—review and editing; Joan Chung, Yoo Lim Lee, Chen Liu, Emily Yang, Zhaoxin Wen, Investigation, Writing—review and editing; Andrew G Clark, Conceptualization, Supervision, Funding acquisition, Project administration, Writing—review and editing; Philipp W Messer, Conceptualization, Supervision, Funding acquisition, Writing—original draft, Project administration, Writing—review and editing

### Author ORCIDs

Jackson Champer http://orcid.org/0000-0002-3814-3774
Philipp W Messer https://orcid.org/0000-0001-8453-9377

### Decision letter and Author response

Decision letter https://doi.org/10.7554/eLife.41439.009
Author response https://doi.org/10.7554/eLife.41439.010

## Additional files

### Supplementary files

• Supplementary file 1. Plasmid construction details and oligonucleotide sequences.
DOI: https://doi.org/10.7554/eLife.41439.005
• Supplementary file 2. Fly phenotype data and rate calculations.
DOI: https://doi.org/10.7554/eLife.41439.006
• Transparent reporting form
DOI: https://doi.org/10.7554/eLife.41439.007

### Data availability

All data generated are available in Supplementary file 2.

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
