## [Decision Letter]

Thank you for submitting your article "Molecular safeguarding of CRISPR gene drive experiments" for consideration by *eLife*. Your article has been reviewed by two peer reviewers, and the evaluation has been overseen by a guest Reviewing Editor and Patricia Wittkopp as the Senior Editor. The following individual involved in review of your submission has agreed to reveal his identity: Owain Edwards (Reviewer #1). Reviewer #2 remains anonymous.

The reviewers have discussed the reviews with one another and the Reviewing Editor has drafted this decision to help you prepare a revised submission.

Summary:

The major concern with the manuscript is one of clarity. The title and Abstract are somewhat misleading, as they suggest that the paper investigates how good the synthetic target and split drive strategies achieve their safeguarding aims, when in actual fact the paper is focused on how well they perform relative to standard gene drive designs. Nor does the Abstract mention the novel findings on maternal deposition. Moreover, there is little background available to orient the nonspecialist reader. The reviewers recommend the following general steps:

Essential revisions:

Purpose: The Abstract and Introduction of the manuscript should clearly state the purpose of the study – to assess whether two proposed methods of safeguarding gene drive research in the laboratory, synthetic target drives and split drives, will produce drive constructs that function in a manner comparable to earlier gene drive constructs that did not use these safeguards. To avoid misleading the reader into thinking that multiple-generation drive experiments were performed or that it was about testing the efficacy of the safeguards, it should specify that relevant drive parameters were measured, such as homing frequency, resistance allele formation, parental deposition frequency, etc. One sentence of the Abstract should mention the interesting results from maternally-deposited vs zygotically-expressed Cas9 and gRNA.

Context 1: An additional paragraph in the Introduction should clearly summarize all previous publications, including relevant preprints, involving CRISPR-based gene drive in insects. This should clearly detail the discrepancies between *Drosophila* and mosquitoes, including male recombination frequencies.

Context 2: The Introduction should clearly describe synthetic site targeting and split drive, including the amount of additional time or research steps required. For example, synthetic site targeting requires an additional transgenesis step, while split drive may require one fewer step if a pre-existing nuclease-expressing line is used. Also pertinent is their relevance to the target application. For example, when contextualizing synthetic site targeting, we recommend noting that moving a target gene from a pest species into a model organism (thereby creating a synthetic site) could permit researchers to test drive systems intended to affect the pest species in the model organism. Notably, a recently published PNAS article (Oberhofer et al., 2018) would have been safer and more relevant had the target genes been moved from the pest D. suzukii into *D. melanogaster*, so raising awareness of this point may be useful.

Results: We recommend reporting the results on split drive in two separate sections. The first section should compare the performance of the split drive constructs relative to past experiments that did not use these safeguards, including in the authors' lab, to address the primary topic of the study. Only once this is completely resolved should the second section, which may deserve its own heading, detail the novel findings regarding maternal deposition vs zygotic expression. Particularly noteworthy is the lack of resistance allele formation when Cas9 is maternally deposited into an embryo that zygotically expresses gRNA.

Perhaps the most surprising result is that "even in individuals that lack any genomic source of Cas9, maternally-deposited Cas9 can persist through to gametocyte formation in the germline, where it can then facilitate successful drive conversion." That this occurs at efficiency comparable to that of organisms that do inherit a genomic source of Cas9 is so surprising that we highly recommend the authors quickly repeat this specific cross to verify the finding.

Statistics: Because these experiments were independent of earlier studies that did not use safeguards, statistical comparisons of the conversion efficiencies may need to be justified by some statements that the designs were identical with respect to any experimental design parameters that could reasonably affect the dependent variables measured.

Language: Given the high-profile nature of the field, we advise avoiding generalities and overstatement throughout the manuscript.

---

## [Author Response]

Essential revisions:Purpose: The Abstract and Introduction of the manuscript should clearly state the purpose of the study – to assess whether two proposed methods of safeguarding gene drive research in the laboratory, synthetic target drives and split drives, will produce drive constructs that function in a manner comparable to earlier gene drive constructs that did not use these safeguards. To avoid misleading the reader into thinking that multiple-generation drive experiments were performed or that it was about testing the efficacy of the safeguards, it should specify that relevant drive parameters were measured, such as homing frequency, resistance allele formation, parental deposition frequency, etc. One sentence of the Abstract should mention the interesting results from maternally-deposited vs zygotically-expressed Cas9 and gRNA.

We have revised and expanded the Abstract to make the purpose of our experiments clearer and now also mention explicitly our new mechanistic findings.

Context 1: An additional paragraph in the Introduction should clearly summarize all previous publications, including relevant preprints, involving CRISPR-based gene drive in insects. This should clearly detail the discrepancies between Drosophila and mosquitoes, including male recombination frequencies.

We thank the reviewers for this suggestion and have added a paragraph summarizing previous work on CRISPR homing gene drive systems, which also provides citations to the relevant literature.

Context 2: The Introduction should clearly describe synthetic site targeting and split drive, including the amount of additional time or research steps required. For example, synthetic site targeting requires an additional transgenesis step, while split drive may require one fewer step if a pre-existing nuclease-expressing line is used. Also pertinent is their relevance to the target application. For example, when contextualizing synthetic site targeting, we recommend noting that moving a target gene from a pest species into a model organism (thereby creating a synthetic site) could permit researchers to test drive systems intended to affect the pest species in the model organism. Notably, a recently published PNAS article (Oberhofer et al., 2018) would have been safer and more relevant had the target genes been moved from the pest D. suzukii into D. melanogaster, so raising awareness of this point may be useful.

This section has been considerably expanded in our revision, following the reviewers’ suggestions. We now discuss explicitly the additional effort required in the engineering of these systems, but also potential advantages of such systems in addition to confinement.

Results: We recommend reporting the results on split drive in two separate sections. The first section should compare the performance of the split drive constructs relative to past experiments that did not use these safeguards, including in the authors' lab, to address the primary topic of the study. Only once this is completely resolved should the second section, which may deserve its own heading, detail the novel findings regarding maternal deposition vs zygotic expression. Particularly noteworthy is the lack of resistance allele formation when Cas9 is maternally deposited into an embryo that zygotically expresses gRNA.

We have broken the split drive section into two parts, as suggested.

Perhaps the most surprising result is that "even in individuals that lack any genomic source of Cas9, maternally-deposited Cas9 can persist through to gametocyte formation in the germline, where it can then facilitate successful drive conversion." That this occurs at efficiency comparable to that of organisms that do inherit a genomic source of Cas9 is so surprising that we highly recommend the authors quickly repeat this specific cross to verify the finding.

We were able to expand the sample size for these experiments, confirming our previous findings. This new data is now also incorporated into the manuscript. However, we want to note that even though drive conversion efficacy in crosses relying on maternal Cas9 was comparable to previous drives, it still was lower than for a “complete” split drive (which was actually higher than most previous *Drosophila* homing drives, as discussed).

Statistics: Because these experiments were independent of earlier studies that did not use safeguards, statistical comparisons of the conversion efficiencies may need to be justified by some statements that the designs were identical with respect to any experimental design parameters that could reasonably affect the dependent variables measured.

We added a note in the main manuscript and in Supplementary file 2, dataset S4 describing the similarities between the drives and mentioning that other experimental parameters were identical.